# The Effects of Breastfeeding for Four Months on Thinness, Overweight, and Obesity in Children Aged 3 to 6 Years: A Retrospective Cohort Study from National Physical Fitness Surveillance of Jiangsu Province, China

**DOI:** 10.3390/nu14194154

**Published:** 2022-10-06

**Authors:** Huiming Huang, Yanan Gao, Na Zhu, Guoqing Yuan, Xiaohan Li, Yong Feng, Linna Gao, Junwu Yu

**Affiliations:** 1Faculty of Sports Science, Research Academy of Grand Health, Ningbo University, Ningbo 315211, China; 2Jiangsu Research Institute of Sports Science, Nanjing 210034, China; 3Ningbo College of Health Sciences, Ningbo 315099, China

**Keywords:** breastfeeding, thinness, overweight, obesity, effect, retrospective cohort study

## Abstract

Objective: To explore the effects of breastfeeding during the first four months of life on thinness, overweight, and obesity and to analyze the influential factors in children aged three to six years in eastern China. Methods: This study was designed as a retrospective cohort study, and the Strengthening the Reporting of Observational Studies in Epidemiology (STROBE) guidelines were followed. A total of 8053 subjects were included in this secondary analysis of data from the 2015 “Physical Fitness Surveillance data of Jiangsu, China”. The subjects were classified into three groups on the basis of feeding patterns: breastfeeding, mixed feeding, and formula feeding. The International Obesity Task Force (IOTF) definitions of BMI were used to define thinness, overweight, and obesity. Multivariate logistic regression models and subgroup analysis were used to assess the association between feeding patterns and childhood thinness, overweight, obesity, and overweight/obesity, adjusted for potential confounders (sex, age grade, area, region/economy, gestational age, birthweight, childbearing age, mother’s education, and caretaker). Results: The prevalence of breastfeeding was 63.8%, and the prevalence of thinness, overweight, obesity, and overweight/obesity reached 2.7%, 11.2%, 4.7%, and 15.9%, respectively. Breastfeeding participants had a lower risk of overweight and overweight/obesity with adjusted ORs of 0.652 (95% CI: 0.533, 0.797; *p* < 0.001) and 0.721 (95% CI: 0.602, 0.862; *p* < 0.001), respectively; however, there was no difference in thinness and obesity (both *p* > 0.05) compared with formula feeding. There was no statistical difference between mixed and formula feeding, in terms of thinness, overweight, obesity, or overweight/obesity (all *p* > 0.05). Subgroup analysis showed that breastfeeding for three years, preterm, and a childbearing age of 25–29 years had higher adjusted ORs for thinness, and in 5–6 years, urban areas, southern/developed economy regions, post-mature, childbearing age ≥ 25 years, and other caretakers had higher and invalid breastfeeding-adjusted ORs (all *p* > 0.05 except overweight in the urban grade) for both overweight and overweight/obesity. Conclusions: Breastfeeding during the first four months was not associated with the thinness of children aged 3–6 years in eastern China, and the protective effect of breastfeeding against overweight or overweight/obesity could be confirmed. However, the effects of breastfeeding on thinness, overweight, and obesity may change or become invalid in some subgroups, suggesting that there may be potential interactions between feeding patterns and influential factors.

## 1. Introduction

Over the past few decades, a dramatic increase in the prevalence of overweight and obesity, not only among adults, but also among children has been documented in many countries [1,2,3]. This has caused concern among public health practitioners because childhood obesity increases the risk of obesity in adulthood [3,4,5]. Excessive weight gain makes children especially vulnerable to noncommunicable diseases such as diabetes and cardiovascular disease at a young age, and they remain obese into adulthood [6]. Screening studies on the nutritional status of children and adolescents in many populations around the world have shown that obesity is accompanied by the other and often overlooked problem of underweight or thinness, which also has a negative impact on population health [7,8]. Malnutrition is associated with both under-nutrition and over-nutrition, which causes the body to get an improper amount of nutrients to maintain tissues and organ function. Under-nutrition is caused by insufficient food intake, poor utilization of nutrients due to illnesses, or a combination of these factors. Obesity in children is currently a global health challenge that contributes to the “double burden” of malnutrition, which afflicts children in developing countries, emphasizing the need for strategies to prevent obesity [9].

There has been some evidence that early-life exposures can increase the risk of obesity [10]. Among other factors, breastfeeding has been hypothesized as a potential priming factor against overweight or obesity. Over the past 20 years, many relevant studies, including prospective and cross-sectional studies [11,12,13,14], systematic reviews, and meta-analyses [15,16,17], have supported the potential value of breastfeeding as a protective factor in reducing the risk of childhood obesity. However, a strong, clear, and consistent body of evidence shows that breastfeeding causally reducing the risk of overweight or obesity is unwarranted [18,19]. Studies disputing the protective role of breastfeeding against obesity are based primarily on uncontrolled confounding and selection bias [18,19]. Studies from Hong Kong, Sweden, Belarus, as well as a recent study in six European countries (Belgium, Bulgaria, Germany, Greece, Poland, and Spain) showed that there was no relationship between breastfeeding and BMI, overweight, and obesity after adjusting for sex, birth weight, gestational age, social class, exposure to passive smoking, parity, maternal age at birth, location of birth, as well as critical infant morbidity [20,21,22,23]. The authors then concluded that breastfeeding was not related to pediatric obesity in their claim, insisting that the previously observed protective effects may be a result of socially determined confounders such as social class, maternal obesity, and maternal smoking [20,21,22,23].

Random intervention in breastfeeding is not feasible for practical reasons, and thus the current original articles are observational studies, such as retrospective or prospective cohort studies and cross-sectional studies. This means that confounding factors may be a crucial point of the effect of breastfeeding and that some uncertain factors may weaken or even eliminate that effect; however, limited potential confounders were included in different studies, which resulted in increased uncertainty of the results. In addition, the value of subgroup analysis has been overlooked in previous studies. Subgroup analysis in previous systematic reviews mainly focused on feeding duration, study design, and different races [15,16,17]. Whether breastfeeding is effective protect against obesity or not, the effect of breastfeeding may be varied in different subgroups. Does the effect of breastfeeding on overweight and obesity diminish as children get older? Do the effects differ in urban and rural areas? What about different gestational ages, different economic regions, and different child caretakers? Subgroup analyses may provide not only a more precise and specific assessment but also a research foundation and epidemiological value for the potential interactions between feeding practices and other child obesity-related factors.

Notably, breastfeeding has shown a protective role against overweight and obesity, so does breastfeeding increase the risk of under-nutrition? Unfortunately, research on the relationship between breastfeeding and underweight or thinness is still limited. Therefore, we conducted this secondary analysis using well-controlled Physical Fitness Surveillance data of Jiangsu Province in China to (a) evaluate the relationship between breastfeeding for four months and thinness to fill in the current knowledge gap; (b) determine whether breastfeeding, after taking into account several potential confounders not included in previous studies, still has an independent protective effect against overweight and obesity; and (c) determine the influencing factors on the relationship between breastfeeding for four months and overweight/obesity through multiple subgroup analysis.

## 2. Materials and Methods

### 2.1. Study Design

This study was designed as a retrospective cohort study. The research techniques adhere to the STROBE (Strengthening the Reporting of Observational Studies in Epidemiology) Statement [24].

### 2.2. Participants and Data Collection

We used the children’s data from the 2015 “Physical Fitness Surveillance data of Jiangsu, China” (PFSJ), a work item of the sports department of the government. PFSJ selected a representative sample of the civilian, noninstitutionalized Chinese population using a complex, stratified, multistage probability and conducted a series of surveys from 2000 to monitor the physical health [25]. The first stage covered all 13 prefecture-level cities in Jiangsu Province. Subsequently, the data included two districts (for urban area) and two counties (for rural area) from each city (second stage), two streets/towns from each district/county (third stage), and two neighborhood committees/villages from each street/town (fourth stage). The final stage used systematic sampling to select kindergartens and ensured that the minimum sample of children aged 3–6 years in each age group was not less than 35. Finally, data were collected from 56 kindergartens in 13 prefecture-level cities (Nanjing, Suzhou, Wuxi, Changzhou, Nantong, Zhengjiang, Yangzhou, Taizhou, Xuzhou, Huaian, Suqian, Lianyungang, and Yancheng). Written informed consent was obtained from each participant’s parents. This secondary analysis does not require ethical review, and the study design was approved by the ethical review committee of the Faculty of Sports Science of Ningbo University’s Institutional Ethics Board (NO. TY2021001).

The data of PFSJ, including physical fitness tests and questionnaires, were checked by a supervisor to control quality and then reported to the Physical Fitness Surveillance Center; thus, we obtained complete and high-quality data for secondary analysis. We extracted information on each child’s height, weight, feeding patterns, and other factors. Finally, 8053 subjects were included in our analysis after deleting 5 samples with missing feeding patterns.

### 2.3. Measurements and Definition

#### 2.3.1. Outcomes or Dependent Variables

The dependent variables in this study were thinness, overweight, and obesity. Measurement was based on the “National Physical Fitness Evaluation Standard” [25]. Height was measured using a stadiometer while participants were standing upright and without shoes. Their weight was measured using a standard digital scale. Height and weight were recorded to the nearest 0.1 cm and 0.1 kg. Body Mass Index (BMI): BMI was calculated as BMI = $\mathrm{Weight}\left(\mathrm{kg}\right)/{\left(\mathrm{Height}\right)}^{2}\left(m\right)$. Definition: the International Obesity Task Force (IOTF) defines overweight and obesity with cut-offs of BMI 25 and 30 at 18 years, respectively, tracked back along the centile curves to age two years [26]. The resulting sets of values, from 2 to 18 years of age at six-month intervals, were defined as the cutoff thresholds for overweight and obesity in children in 2000 [26], and we determined the cut-offs of thinness in children and adolescents, based on BMI 17 at 18 years in 2007 (equates to grade 2 thinness in adults using the WHO’s definition) [27].

#### 2.3.2. Feeding Patterns and Covariates

Feeding patterns and nine potential childhood obesity-related factors were included as independent variables and covariates in this study and were collected through a self-report questionnaire. Feeding patterns were divided into three categories: breastfeeding was defined as exclusive breastfeeding during the first four months of life without any additional foods or fluids; formula feeding was defined as artificial feeding with formula milk or other additional foods and never breastfeeding; mixed feeding was defined as a combination of breastfeeding and formula feeding.

Covariates included sex, age grade (3 years, 4 years, 5 years, 6 years), area (urban and rural), region/economy (south and north; all cities with a GDP more than one trillion RMB are in southern Jiangsu, and thus south is equal to the economically developed region and north is the less-developed region), gestational age (recorded from each child’s birth certificate; preterm ≤36 weeks, mature 37–41 weeks, post-mature ≥42 weeks), birthweight (recorded from each child’s birth certificate by his/her parents; low was defined as birthweight less than the 10th percentile of the birthweight reference for China [28]), childbearing age (<25 years, mature 25–29 years, ≥30 years), mother’s education (low is defined as qualification less than high school; medium is equal to high school or vocational school, and high means a bachelor’s degree or above), and caretaker (primary caregiver of children before the age of 3; parents, grandparents, and others).

### 2.4. Statistical Analysis

All analyses were carried out in SPSS 26.0. Categorical variables were reported as frequencies with proportions, and continuous variables were reported as means. The prevalance of categorical variables and means were compared using analysis of x^2^ and variance (ANOVA), respectively. Multivariate logistic regression models were used to evaluate the associations between feeding patterns and each child’s thinness, overweight, obesity, and overweight/obesity (the above outcomes were divided into dichotomous variables). The odds ratios (ORs) and 95% confidence interval were used for outcomes adjusted for potential confounders (sex, age grade, area, region/economy, gestational age, birthweight, childbearing age, mother’s education, and caretaker). A *p*-value < 0.05 was considered statistically significant and all tests were two-sided; subgroup analysis was performed based on the above characteristic information.

Sensitivity analyses were conducted using an alternative cut-point for the definition of thinness, overweight, and obesity (grade 1 thinness based on the IOTF definition: centile curves were drawn to pass through the cut-off of BMI 18.5 at 18 years; overweight, obesity of Chinese researchers’ definition: the cut-off of BMI 24 and 28, respectively, at 18 years) (Appendix A).

## 3. Results

### 3.1. Participants’ Characteristics and Univariate Analysis on Thinness, Overweight, and Obesity

A total of 8053 participants (4022 males and 4031 females) aged three to six years were included in this study, among which the prevalence of thinness, overweight, obesity, and overweight/obesity reached 2.7%, 11.2%, 4.7%, and 15.9%, respectively. The total percentage of breastfeeding for four months was 63.8%, while the percentage of mixed feeding and formula feeding reached 20.3% and 15.9%, respectively. A minor proportion of the records on covariates was not collected. The missing values were ignored as follows: gestational age: 9 (0.1%); birthweight: 137 (1.7%); childbearing age: 54 (0.6%); mother’s education: 6 (0.1%); caretaker: 7 (0.1%).

Table 1 shows the characteristics of the participants and univariate analysis of the prevalence of thinness, overweight, obesity, and overweight/obesity. A Chi-square test was used for group comparisons of categorical variables. Univariate analysis showed that the differences in the prevalence of thinness were statistically significant in terms of sex, age grade, area, gestational age, birthweight, and childbearing (*p* < 0.05) and revealed that participants who were female (3.2%), age three years (4.0%), rural (3.4%), preterm (4.5%), low birthweight (4.2%), and childbearing age < 25 years (3.0%) may have higher thinness prevalence. By comparison, the following participants had a higher overweight prevalence (*p* < 0.05): formula feeding (13.6%), male (11.9%), urban (12.9%), north/less-developed economy (13.4%), normal birthweight (11.8%), and medium qualification of the mother (12.4%), and those with mixed feeding (5.9%), male (5.5%), age six years (6.0%), urban (5.7%), and north/less-developed economy (5.9%) had a higher obesity prevalence (*p* < 0.05). The differences in overweight/obesity prevalence showed a similar trend to that of overweight.

### 3.2. The Prevalence of Breastfeeding in Subgroups of Covariates

The overall prevalence of breastfeeding during the first four months of life was 63.8% in this study. The distribution of breastfeeding prevalence was different among several subgroups. The data were analyzed by age stratification (3–4 years old and 5–6 years old, Figure 1).

Univariate analysis (Chi-square test) showed that the differences of breastfeeding prevalence in the area, gestational age, childbearing age, mother’s education, and caretaker had statistical significances in both the 3–4 year and 5–6 year age group (all *p* < 0.05) and indicated that the following participants had a higher breastfeeding prevalence: thos in rural areas (65.9–70.4%), mature (62.1–67.1%), childbearing age <25 years (67.0–68.6%), low mother’s qualification (65.0–68.6%), and parents as caretakers (65.5–69.0%) (Figure 1). In general, the breastfeeding prevalence of the 5–6-year-old group was higher than that of the 3–4-year-old group.

### 3.3. Mean Distribution of BMI in Subgroups of Covariates

The mean distribution of BMI in subgroups of covariates showed differences. Univariate analysis (one-way ANOVA) showed that the differences in BMI means were statistically significant in terms of the feeding patterns, age grade, area, region, birthweight, and mother’s education (all *p* < 0.05) and indicated that participants with mixed feeding (16.18%), age six years (16.11%), in urban areas (16.21%), in the north (16.24%), with normal birthweight (16.09%), and a high mother’s qualification (16.10%) had higher BMIs (Figure 2).

### 3.4. Association among Breastfeeding, Thinness, Overweight, and Obesity in Multivariate Logistic Regression Models

The associations among breastfeeding, thinness, overweight, and obesity are shown in Table 2. Multivariate logistic regression models included ten factors (feeding patterns, sex, age grade, area, region/economy, gestational age, birthweight, childbearing age, mother’s education, and caretaker); OR values were calculated based on the reference group after adjusting for the other nine factors.

Based on the adjustment of the potential confounding factors, participants who breastfed for four months had a lower risk of being overweight and overweight/obesity, with adjusted ORs of 0.652 (95% CI: 0.533, 0.797; *p* < 0.001) and 0.721 (95% CI: 0.602, 0.862; *p* < 0.001), respectively; however, there were no differences in thinness and obesity (both *p* > 0.05) compared to formula feeding. Additionally, irrespective of thinness, overweight, obesity or overweight/obesity, there was no statistical difference between mixed feeding and formula feeding (all *p* > 0.05). Moreover, other factors also had effects on the outcomes. Participants who were female (adjusted OR 1.566 vs. male), aged 4–6 years (adjusted OR 0.444–0.582 vs. 3 years), lived in rural areas (adjusted OR 1.507 vs. urban), had normal birthweight (adjusted OR 0.614 vs. low), and grandparents as caretakers (adjusted OR 0.614 vs. parents) had a statistical difference in thinness (all *p* < 0.05). There were statistical differences in overweight/obesity in females (adjusted OR 0.786 vs. male), those aged 5–6 years (adjusted OR 1.277–1.549 vs. 3 years), participants who lived in rural areas (adjusted OR 0.698 vs. urban), in the north/developing areas (adjusted OR 1.769 vs. south/developed), and had normal birthweight (adjusted OR 1.689 vs. low), with all *p* < 0.05 (Table 2).

Further subgroup analyses of the adjusted ORs of breastfeeding with respect to thinness, overweight, and obesity are shown in Table 3. The subgroup aged three years (adjusted OR 2.373), who was preterm (adjusted OR 8.854), with childbearing age 25–29 years (adjusted OR 2.334) had higher breastfeeding adjusted ORs for thinness, which may indicate a potential interaction between feeding patterns and gestational age and childbearing age with respect to thinness. Moreover, subgroup analysis showed that participants aged 5–6 years, in the urban area, in the south/developed economy region, post-mature, childbearing age ≥25, and other caretakers had higher and invalid breastfeeding-adjusted ORs (all *p* > 0.05 except overweight participants in the urban area) for both overweight and overweight/obesity, which indicated that the effect of protection against overweight and overweight/obesity of breastfeeding did not exist in the above subgroups. It also suggested possible potential interactions between breastfeeding and the above factors in terms of overweight/obesity (Table 3).

### 3.5. Sensitivity Analysis

Sensitivity analyses (grade 1 thinness of IOTF definition: centile curves were drawn to pass through the cut-off of BMI 18.5 at 18 years, and overweight, obesity of Chinese researchers’ definition: cut-off of BMI 24, 28 at 18 years) showed that the adjusted OR of breastfeeding had statistical significance (OR = 1.307, *p* = 0.020), indicating that breastfeeding for four months had a higher risk of thinness than formula feeding when the cutoff point was raised. Although the cut-off points of overweight and obesity had decreased, compared with formula feeding, breastfeeding maintained a consistent trend for overweight and obesity. Additionally, the adjusted OR had no difference in other factors such as gender, area, region, birthweight, childbearing age, mother’s education, and caretaker, except for age and gestational age (Appendix Table A1).

## 4. Discussion

The association between breastfeeding and childhood obesity has long been under debate. Despite a few contradictory reports, the findings of several systematic reviews and meta-analyses support the relationship between breastfeeding and reduced risk of obesity [15,16,17]. The use of uniform potential confounders and a reduction in selection bias would help resolve the controversy over the protective effect of breastfeeding on overweight and obesity [16,17]. We included nine potential confounders in this study, some of which had not previously been reported. In addition, we investigated the relationship between breastfeeding during the first four months of life and thinness and conducted subgroup analysis, potential interaction analysis, and sensitivity analysis, which improved the reliability of our findings. These findings may provide valuable information on the “double burden” of malnutrition in the Chinese population regarding breastfeeding. Interestingly, one study reported that the protective effect of breastfeeding against early childhood overweight and obesity may vary by race and ethnicity [29]. In a systematic review that included 25 studies [16], only two investigated the Chinese population; therefore, more research on the Chinese population is needed to confirm these findings.

In this study, the overall prevalence of exclusive breastfeeding during the first four months of life was 63.8%. Several studies from China, including a large sample study and a review study, have found that the prevalence of exclusive breastfeeding for 4–6 months was 52–83% [30,31]. Therefore, it is credible that we obtained a high prevalence of exclusive breast feeding for four months, because the longer the duration, the lower the breastfeeding prevalence [32]. The definition of exclusive breastfeeding for four months by the China National Physical Fitness Surveillance Center has been used in five waves of nationwide fitness and health surveys for longitudinal comparison since 2000. In addition, the addition of complementary foods after four months of age for Chinese infants is usually recommended; therefore, breastfeeding during the first four months of life is more likely to be accurately recalled by mothers, which helps reduce recall errors. A previous review study also showed that the duration of breastfeeding and the anti-obesity effect are positively correlated [17]; therefore, the duration of breastfeeding more than four months may obtain different or better effects against overweight or obesity.

We used univariate analysis to assess the relationship between feeding patterns and other factors, BMI, and the prevalence of thinness, overweight, and obesity. We found that most of the included factors were likely to be related to thinness, overweight, and obesity. Our findings also showed that the distribution of all factors in breastfeeding prevalence was unbalanced, except sex. In observational studies, the covariates with unbalanced distribution may have an important impact on the conclusion, which may be the source of confounding bias. For example, when Chinese children’s caretakers were grandparents, they were more likely to feed more because the older generation of Chinese had the concept of “fat children are lucky”; meanwhile, the breastfeeding prevalence of children whose caretakers were grandparents was lower than that of children whose caretakers were parents, implying that caretakers should be considered a confounding factor when analyzing the relationship between breastfeeding and obesity in the Chinese population. Qiao et al.’s systematic review included 26 studies analyzing the relationship between breastfeeding and obesity, with the main adjusted factors for each study including 3–6 of the following factors: gender, ethnicity/race, child’s age, gestational age, gestational weight gain, birth weight, rapid infant weight gain, timing of introduction of solids, mother’s BMI, mother’s education, family income, and mother’s occupation [17]. It has been suggested that several confounding variables required special attention, such as social economic status and birth weight [22,33]. As a result, we adjusted for more factors, including area (urban or rural), region (south/developed economy or north/less-developed economy), childbearing age, and caretaker, which could provide more evidence for the effectiveness of breastfeeding.

Much has been written about the epidemic of child obesity, but malnutrition in infants, children, and adolescents also poses a considerable international public health problem [34,35]. We adopted the ITOF definition and criteria for “thinness” (centile curves pass through the cut-off of BMI 17 at 18 years), which means low BMI for age and avoids potential confusion between the terms “wasting” and “underweight“ in children [27]. We reported a prevalence of thinness of 2.7% in Jiangsu Province, China, which is lower than the global average [36]. We found no statistical difference in the prevalence of thinness among breastfeeding, mixed feeding, and formula feeding in children aged 3–6 years after adjusting for confounding factors, suggesting that when beneficial health outcomes were obtained, breastfeeding did not increase the risk of under-nutrition due to thinness. The same results were found in a study of indigenous Ecuadorians younger than two years, showing that breastfeeding was not associated with stunting or underweight [37]. However, the results of the subgroup analysis showed that participants aged three years, who were preterm, and a childbearing age of 25–29 years had higher adjusted ORs for thinness, indicating a potential interaction of feeding patterns, gestational age, and childbearing age on thinness; particularly in the preterm grade, the adjusted OR reached 8.854 and was statistically significant, indicating that the interaction was significant—the risk of thinness was additionally added when breastfeeding and preterm coexisted. However, the effect of breastfeeding being reduced or removed in older age grades could also be due to a washout effect in older children. The epidemiological value of this interaction is that weight gain should still be recommended for specific groups to prevent under-nutrition. Furthermore, in the sensitivity analysis, a weak positive association between breastfeeding and slight thinness was found when the thinness cut point of the IOTF definition was increased to BMI 18.5 at 18 years.

Although there was no effect on obesity alone, breastfeeding demonstrated a stronger effect on the prevalence of overweight and overweight/obesity among children aged 3–6 years, with adjusted ORs of 0.652 for overweight and 0.721 for overweight/obesity (formula feeding as reference), after adjusting for nine factors. This could imply that the protective effect of breastfeeding is mainly reflected in overweight.

Our findings support the protective effect of breastfeeding against overweight and obesity. Furthermore, we discovered that mixed feeding had no such effect on thinness, overweight, obesity, or overweight/obesity. Our findings are similar to previous meta-analyses [16,38], which reported ORs of 0.83–0.87 for the risk of obesity in ever-breastfed preschoolers compared to never-breastfed preschoolers. Some hypothesized mechanisms that could link breastfeeding to childhood obesity are listed below. The first possible mechanism is about the behavioral justifications. While formula feeding can teach the baby to neglect satiety cues, breastfeeding can help the infant gain control of his or her food intake [38]. Second, studies have shown that breastfed infants are more likely to delay solid food intake, which decreases the risk of childhood obesity [39]. Another possible mechanism is the nutritional explanation. Breastmilk contains less protein than infant formula. When Weber et al. reduced the protein content of infant formula to levels comparable to those found in breast milk, they discovered that the BMI of breastfed infants at age six was comparable to that of formula infants [40]. Furthermore, the taste of the diet from the mother transmitted through breast milk may influence the infant’s taste and food acceptance later in life. Breastfed infants are less likely to develop picky eating habits, which leads to a healthier diet that contributes to non-obesity [41].

The further subgroup analyses revealed that although the overall effect was strong and adjusted ORs were below 1 for most subgroups, the effect of protection against overweight and overweight/obesity of breastfeeding in children aged 5–6 years, participants living in urban areas and/or the south/developed economy region, post-mature, a childbearing age of ≥25, and other caretakers was weakened or eliminated. Likewise, it also suggested a potential interaction between breastfeeding and the above factors on overweight/obesity. Subgroup analyses in previous studies have focused on the duration of breastfeeding [42,43,44], and thus a dose–response relationship between breastfeeding duration and childhood obesity has been confirmed; the longer the duration of breastfeeding, the more pronounced the effect. However, the analyses of other subgroups in previous studies were not extensive or conclusive.

This study has several strengths. First, the data for this study were collected from an equal-volume sample of 13 cities in Jiangsu with a well-representative demographic profile, and all measures and covariates were recorded with strict quality supervision, which reduced the selective bias of the sample. Second, the study fills the research gap by exploring the relationship between breastfeeding and thinness in the Chinese population. Third, nine confounding factors were included in this study, some of which have not been reported in previous studies. Fourth, an extensive subgroup analysis was performed in this study, which not only made the findings more comprehensive and accurate, but also explored the potential or possible interactions. Finally, in this retrospective cohort study, the inclusions of feeding patterns and other confounding factors definitely occurred prior to the outcome variables, and therefore, causality conclusions could be drawn from the results, which are superior to cross-sectional studies in terms of the level of evidence.

Some limitations should be recognized. Firstly, this study was based on retrospective self-reported data (3–6 years of breastfeeding history); therefore, recall bias may exist. Secondly, unmeasured confounding factors may potentially bias the results. Finally, the data were collected from eastern China, and so the subjects cannot represent the national Chinese population.

## 5. Conclusions

Breastfeeding was not associated with thinness in children aged 3–6 years in eastern China, and the protective effect of breastfeeding against overweight or overweight/obesity was confirmed. However, the effects of breastfeeding on thinness, overweight, and obesity may change or become invalid in some subgroups, suggesting that there may be potential interactions between feeding patterns and influential factors.

## Figures and Tables

**Figure 1 nutrients-14-04154-f001:**
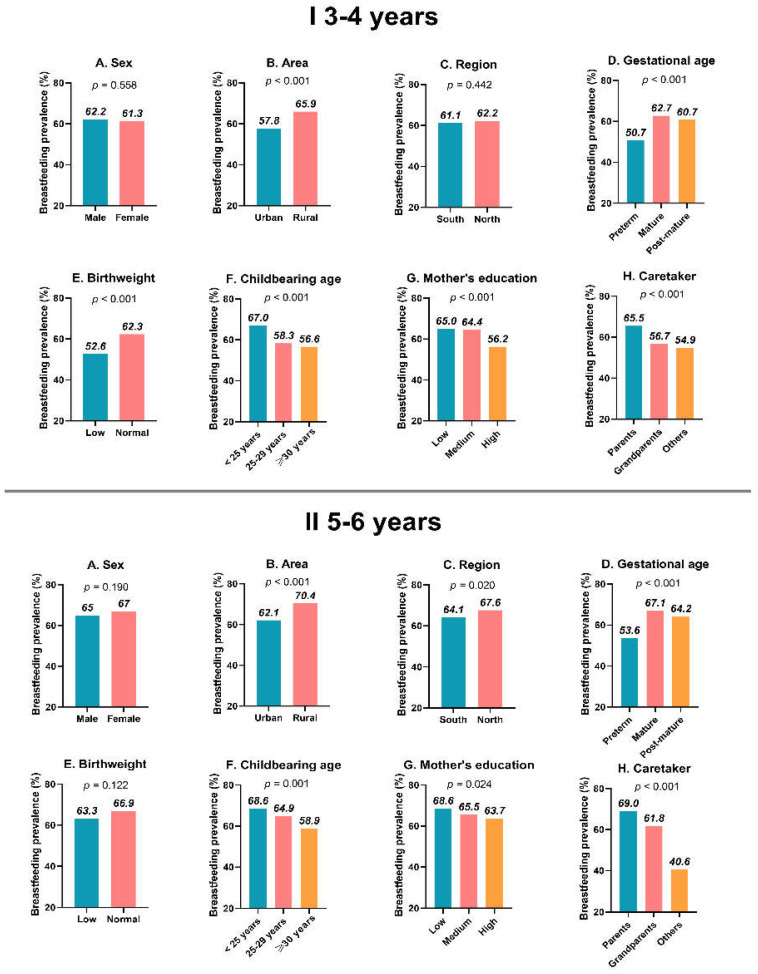
The prevalence of breastfeeding in subgroups. (**I**) 3–4 years; (**II**) 5–6 years.

**Figure 2 nutrients-14-04154-f002:**
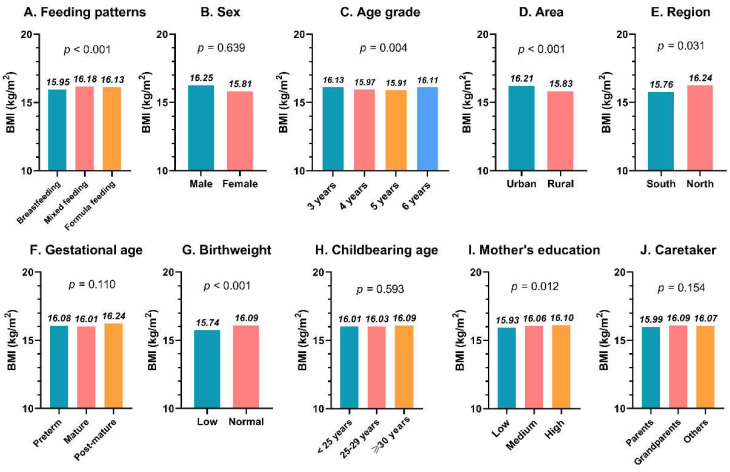
Mean distribution of BMI in subgroups. (**A**) Feeding patterns; (**B**) Sex; (**C**) Age grade; (**D**) Area; (**E**) Region; (**F**) Gestational age; (**G**) Birthweight; (**H**) Childbearing age; (**I**) Mother’s education; (**J**) Caretaker. Abbreviations: BMI, body mass index.

**Table 1 nutrients-14-04154-t001:** Participants’ characteristics and univariate analysis of thinness, overweight, and obesity.

Characteristics	Total *n* (%)	Thinness *n* (%)	*p*	Overweight *n* (%)	*p*	Obesity *n* (%)	*p*	Overweight /Obesity *n* (%)	*p*
Feeding patterns			0.066		<0.001		0.042		<0.001
breastfeeding	5140 (63.8)	155 (3.0)		522 (10.2)		229 (4.5)		751 (14.6)	
mixed feeding	1638 (20.3)	33 (2.0)		203 (12.4)		97 (5.9)		300 (18.3)	
formula feeding	1275 (15.9)	30 (2.4)		174 (13.6)		56 (4.4)		230 (18.0)	
Sex			0.003		0.028		0.001		<0.001
male	4022 (49.9)	87 (2.2)		480 (11.9)		223 (5.5)		703 (17.5)	
female	4031 (51.1)	131 (3.2)		419 (10.4)		159 (3.9)		578 (14.3)	
Age grade			0.001		0.21		0.005		<0.001
3 years	2067 (25.7)	82 (4.0)		205 (9.9)		87 (4.2)		292 (14.1)	
4 years	2040 (25.3)	48 (2.4)		214 (10.5)		76 (3.7)		290 (14.2)	
5 years	2149 (26.7)	51 (2.4)		249 (11.6)		112 (5.2)		361 (16.8)	
6 years	1797 (22.3)	37 (2.1)		231 (12.9)		107 (6.0)		338 (18.8)	
Area			0.001		<0.001		<0.001		<0.001
urban	4219 (52.4)	89 (2.1)		545 (12.9)		241 (5.7)		786 (18.6)	
rural	3843 (47.6)	129 (3.4)		354 (9.2)		141 (3.7)		495 (12.9)	
Region/economy ^a^			0.093		<0.001		<0.001		<0.001
south/developed	3468 (44.4)	109 (3.0)		300 (8.4)		119 (3.3)		419 (11.7)	
north/less-developed	4476 (55.6)	109 (2.4)		599 (13.4)		263 (5.9)		862 (19.3)	
Gestational age			0.017		0.137		0.328		0.133
preterm	538 (6.7)	24 (4.5)		56 (10.4)		32 (5.9)		88 (16.4)	
mature	7087 (88.1)	187 (2.6)		784 (11.1)		327 (4.6)		1111 (15.7)	
post-mature	419 (5.2)	7 (1.7)		59 (14.1)		22 (5.3)		81 (19.3)	
Birthweight ^b^			<0.001		<0.001		0.153		<0.001
low	1045 (13.2)	44 (4.2)		73 (7.0)		40 (3.8)		113 (10.8)	
normal	6871 (86.8)	165 (2.4)		811 (11.8)		337 (4.9)		1148 (16.7)	
Childbearing age			0.036		0.774		0.771		0.785
<25 years	3553 (44.4)	106 (3.0)		394 (11.1)		161 (4.5)		555 (15.6)	
25–29 years	3472 (43.4)	87 (2.5)		382 (11.0)		170 (4.9)		552 (15.9)	
≥30 years	974 (12.2)	23 (2.4)		115 (11.8)		46 (4.7)		161 (16.5)	
Mother’s education ^c^			0.804	66	0.027		0.285		0.025
low	2748 (34.1)	79 (2.9)		278 (10.1)		119 (4.3)		397 (14.4)	
medium	2627 (32.6)	69 (2.6)		326 (12.4)		123 (4.7)		449 (17.1)	
high	2672 (33.2)	70 (2.6)		294 (11.0)		140 (5.2)		434 (16.2)	
Caretaker			0.215		0.180		0.058		0.680
parents	5000 (62.1)	146 (2.9)		583 (11.7)		217 (4.3)		800 (16.0)	
grandparents	2838 (35.3)	65 (2.3)		292 (10.3)		151 (5.3)		443 (15.6)	
others	208 (2.6)	7 (3.4)		23 (11.1)		14 (6.7)		37 (17.8)	

Values are expressed as *n* (%), and the Chi-square test was used for group comparisons of categorical variables. ^a^ South is equal to the economically developed region and north means less-developed; ^b^, Low is defined as birthweight less than the 10th percentile of the birthweight reference for China; ^c^, Low is defined as qualification less than high school, medium is equal to high school or vocational school, and high means bachelor’s degree or above. The missing values are as follows: Gestational age: 9 (0.1%); Birthweight: 137 (1.7%); Childbearing age: 54 (0.6%); Mother’s education: 6 (0.1%); Caretaker: 7 (0.1%).

**Table 2 nutrients-14-04154-t002:** Adjusted ORs (95% CI) for breastfeeding and covariates for thinness, overweight, and obesity in multivariate logistic regression models.

Category	Thinness	*p*	Overweight	*p*	Obesity	*p*	Overweight/ Obesity	*p*
Feeding patterns								
formula feeding	Ref		Ref		Ref		Ref	
mixed feeding	0.682 (0.392, 1.186)	0.175	0.842 (0.665, 1.064)	0.150	1.330 (0.918, 1.926)	0.132	0.960 (0.780, 1.181)	0.698
breastfeeding	1.179 (0.772, 1.801)	0.446	0.652 (0.533, 0.797)	<0.001	1.016 (0.729, 1.415)	0.926	0.721 (0.602, 0.862)	<0.001
Sex								
male	Ref		Ref		Ref		Ref	
female	1.566 (1.159, 2.117)	0.004	0.867 (0.747, 1.006)	0.061	0.675 (0.538, 0.846)	0.001	0.786 (0.691, 0.895)	<0.001
Age grade								
3 years	Ref		Ref		Ref		Ref	
4 years	0.582 (0.396, 0.857)	0.006	1.066 (0.859, 1.324)	0.562	0.807 (0.571, 1.140)	0.223	0.985 (0.815, 1.191)	0.875
5 years	0.497 (0.334, 0.742)	0.001	1.202 (0.972, 1.486)	0.089	1.367 (1.004, 1.861)	0.047	1.277 (1.064, 1.533)	0.008
6 years	0.444 (0.287, 0.687)	<0.001	1.449 (1.167, 1.799)	0.001	1.589 (1.159, 2.177)	0.004	1.549 (1.285, 1.866)	<0.001
Area								
urban	Ref		Ref		Ref		Ref	
rural	1.507 (1.091, 2.082)	0.013	0.744 (0.631, 0.878)	<0.001	0.661 (0.515, 0.849)	0.001	0.698 (0.605, 0.805)	<0.001
Region/economy ^a^								
south/developed	Ref		Ref		Ref		Ref	
north/less-developed	0.792 (0.588, 1.067)	0.125	1.620 (1.387, 1.892)	<0.001	1.868 (1.474, 2.368)	<0.001	1.769 (1.545, 2.024)	<0.001
Gestational age								
preterm	Ref		Ref		Ref		Ref	
mature	0.633 (0.384, 1.044)	0.073	0.962 (0.697, 1.327)	0.812	0.651 (0.432, 0.982)	0.041	0.823 (0.630, 1.075)	0.153
post-mature	0.409 (0.159, 1.049)	0.063	1.347 (0.879, 2.062)	0.171	0.676 (0.363, 1.257)	0.216	1.085 (0.752, 1.566)	0.662
Birthweight ^b^								
low	Ref		Ref		Ref		Ref	
normal	0.614 (0.418, 0.901)	0.013	1.804 (1.374, 2.368)	<0.001	1.324 (0.918, 1.909)	0.133	1.689 (1.348, 2.116)	<0.001
Childbearing age								
<25 years	Ref		Ref		Ref		Ref	
25–29 years	0.810 (0.584, 1.122)	0.205	0.910 (0.771, 1.074)	0.264	0.991 (0.776, 1.267)	0.994	0.929 (0.805, 1.072)	0.314
≥30 years	0.664 (0.388, 1.136)	0.135	1.008 (0.789, 1.288)	0.950	0.873 (0.591, 1.289)	0.494	0.963 (0.775, 1.195)	0.730
Mother’s education ^c^								
low	Ref		Ref		Ref		Ref	
medium	1.068 (0.732, 1.557)	0.733	1.217 (1.002, 1.478)	0.048	0.882 (0.655, 1.187)	0.407	1.115 (0.941, 1.321)	0.209
high	1.180 (0.794, 1.755)	0.413	1.024 (0.833, 1.260)	0.820	0.968 (0.716, 1.309)	0.834	1.010 (0.845, 1.207)	0.961
Caretaker								
parents	Ref		Ref		Ref		Ref	
grandparents	0.713 (0.514, 0.990)	0.044	0.870 (0.739, 1.023)	0.092	1.354 (1.072, 1.710)	0.011	1.001 (0.871, 1.150)	0.994
others	1.751 (0.793, 3.863)	0.166	0.834 (0.509, 1.368)	0.473	1.487 (0.783, 2.824)	0.225	1.008 (0.996, 1.524)	0.971

Multivariate logistic regression models included ten factors (feeding patterns, sex, age grade, area, region/economy, gestational age, birthweight, childbearing age, mother’s education, and caretaker); OR values were calculated based on the reference group after adjusting for the other nine factors. ^a^, South is equal to the economically developed region and north means less developed; ^b^, Low is defined as birthweight less than the 10th percentile of the birthweight reference for China; ^c^, Low is defined as qualification less than high school, medium is equal to high school or vocational school, and high means bachelor degree or above. Abbreviations: Ref, Reference group.

**Table 3 nutrients-14-04154-t003:** Subgroup analysis of adjusted ORs (95% CI) of breastfeeding for thinness, overweight, and obesity.

Category	Thinness	*p*	Overweight	*p*	Obesity	*p*	Overweight/Obesity	*p*
Total	1.179 (0.772, 1.801)	0.446	0.652 (0.533, 0.797)	<0.001	1.016 (0.729, 1.415)	0.926	0.721 (0.602, 0.862)	<0.001
Sex								
male	0.966 (0.505, 1.848)	0.918	0.621 (0.471, 0.820)	0.001	0.946 (0.614, 1.457)	0.801	0.681 (0.532, 0.871)	0.002
female	1.362 (0.777, 2.390)	0.281	0.697 (0.519, 0.937)	0.017	1.170 (0.694, 1.970)	0.556	0.773 (0.595, 1.005)	0.054
Age grade								
3 years	2.373 (0.991, 5.682)	0.052	0.531 (0.350, 0.804)	0.003	1.290 (0.612, 2.718)	0.503	0.655 (0.452, 0.950)	0.026
4 years	0.685 (0.338, 1.387)	0.293	0.466 (0.320, 0.680)	<0.001	0.618 (0.299, 1.281)	0.196	0.478 (0.338, 0.675)	<0.001
5 years	0.740 (0.310, 1.766)	0.497	0.832 (0.551, 1.256)	0.382	1.763 (0.875, 3.555)	0.113	1.045 (0.723, 1.511)	0.814
6 years	1.687 (0.488, 5.833)	0.408	0.996 (0.641, 1.548)	0.986	0.758 (0.425, 1.354)	0.350	0.892 (0.613, 1.297)	0.550
Area								
urban	1.159 (0.615, 2.181)	0.648	0.761 (0.585, 0.990)	0.042	1.095 (0.721, 1.664)	0.669	0.834 (0.660, 1.054)	0.129
rural	1.130 (0.638, 2.000)	0.675	0.534 (0.391, 0.728)	<0.001	0.959 (0.557, 1.651)	0.881	0.607 (0.460, 0.801)	<0.001
Region/economy ^a^								
south/developed	1.212 (0.668, 2.200)	0.526	0.783 (0.558, 1.097)	0.155	0.888 (0.513, 1.538)	0.672	0.804 (0.598, 1.082)	0.149
north/less-developed	1.096 (0.598, 2.008)	0.768	0.587 (0.457, 0.755)	<0.001	1.098 (0.722, 1.669)	0.662	0.680 (0.542, 0.853)	0.001
Gestational age								
preterm	8.854 (1.096, 71.527)	0.041	0.505 (0.244, 1.047)	0.066	0.564 (0.226, 1.406)	0.219	0.478 (0.259, 0.880)	0.018
mature	0.932 (0.603, 1.440)	0.750	0.645 (0.520, 0.801)	<0.001	1.099 (0.758, 1.593)	0.618	0.730 (0.601, 0.886)	0.001
post-mature ^d^	NA	-	1.058 (0.405, 2.764)	0.909	2.418 (0.277, 21.13)	0.425	1.240 (0.502, 3.062)	0.641
Birthweight ^b^								
low	0.943 (0.396, 2.244)	0.894	0.496 (0.263, 0.934)	0.030	1.070 (0.400, 2.860)	0.893	0.616 (0.356, 1.065)	0.083
normal	1.241 (0.759, 2.027)	0.390	0.670 (0.542, 0.829)	<0.001	1.012 (0.710, 1.441)	0.949	0.734 (0.607, 0.888)	0.001
Childbearing age								
<25 years	0.977 (0.537, 1.779)	0.941	0.548 (0.402, 0.747)	<0.001	1.009 (0.608, 1.673)	0.973	0.630 (0.478, 0.831)	0.001
25–29 years	2.334 (1.043, 5.223)	0.039	0.759 (0.555, 1.038)	0.084	1.013 (0.612, 1.676)	0.959	0.814 (0.617, 1.074)	0.145
≥30 years	0.344 (0.144, 1.036)	0.058	0.700 (0.418, 1.171)	0.174	0.850 (0.331, 2.181)	0.735	0.712 (0.446, 1.138)	0.155
Mother’s education ^c^								
low	1.026 (0.510, 2.062)	0.943	0.641 (0.450, 0.915)	0.014	1.112 (0.624, 1.983)	0.719	0.739 (0.540, 1.013)	0.060
medium	1.454 (0.640, 3.303)	0.371	0.774 (0.550, 1.090)	0.143	0.918 (0.526, 1.604)	0.765	0.795 (0.585, 1.079)	0.141
high	1.090 (0.529, 2.247)	0.815	0.558 (0.392, 0.794)	0.001	1.053 (0.580, 1.913)	0.865	0.646 (0.470, 0.887)	0.007
Caretaker								
parents	0.999 (0.599, 1.668)	0.998	0.645 (0.502, 0.828)	0.001	1.037 (0.667, 1.611)	0.873	0.709 (0.565, 0.890)	0.003
grandparents	1.275 (0.579, 2.808)	0.546	0.588 (0.415, 0.834)	0.003	0.946 (0.553, 1.619)	0.840	0.658 (0.485, 0.893)	0.007
others	2.004 (0.602, 6.674)	0.258	3.235 (0.671, 15.60)	0.144	2.353 (0.352,15.73)	0.377	2.741 (0.809, 9.288)	0.105

Formula feeding was the reference group, and adjusted factors included sex, age grade, area, region/economy, gestational age, birthweight, childbearing age, mother’s education, and caretaker in multivariate logistic regression models; ^a^, South is equal to the economically developed region, and north means less-developed; ^b^, Low is defined as birthweight less than the 10th percentile of the birthweight reference for China; ^c^, Low is defined as qualification less than high school, medium is equal to high school or vocational school, and high means bachelor degree or above. ^d^, NA means not applicable due to zero-size data in the combined formula feeding and post-mature group.

## Data Availability

Data sharing is not applicable to this article as no new data were created or analyzed in this study protocol.

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
