# Peer review of "The Effects of Breastfeeding for Four Months on Thinness, Overweight, and Obesity in Children Aged 3 to 6 Years: A Retrospective Cohort Study from National Physical Fitness Surveillance of Jiangsu Province, China"

_nutrients, 2022, doi:10.3390/nu14194154_

Round 1

Reviewer 1 Report

see attached file

Author Response

Please read the responses letter for details.

Reviewer 2 Report

This was an interesting study on the effects of breastfeeding on BMI in children aged 3-6 years that was able to take into consideration quite a few co-factors.  

It is not clear from the description in the methods if the feeding history was reported by the parents (risk of recall or social desirability bias) or was obtained from previous health records. The >63% rate of exclusive breastfeeding for 4 months makes this somewhat suspect in my mind.  This figure is much higher than the 29% rate I found reported for China for 2019.  

In line 145 the authors state that the definition of breastfeeding was "exclusive breastfeeding without any additional food or fluids within 4 months after birth".  Does this mean exclusive breastfeeding for at least 4 months or does it mean exclusive breastfeeding for unknown duration during the first 4 months.  I assumed for "at least" 4 months.  once again, 63% is very high.  

The Authors do not relate to the finding that breastfeeding appeared to be protective for overweight but not for obesity.  This should be addressed.  Moreover their discussion attributes the absence of an effect in the older ages to "interaction" whereas the possibility of washout or limited duration of effect is certainly possible and has been previously reported.  

I found the presentation of the logistic regression for each variable and then the subgroup analysis looking specifically at the effect of each variable on breastfeeding confusing.  

Author Response

(The authors gave the same response as above.)
